Resilience of political leaders and healthcare organizations during COVID-19

http://orcid.org/0000-0002-7554-1798 Baxi Manmeet Kaur 1 mbaxi@lakeheadu.ca
Philip Joshua 2
http://orcid.org/0000-0002-9741-3463 Mago Vijay 1
1 Department of Computer Science, Lakehead University , Thunder Bay, Ontario , Canada
2 Superior Collegiate and Vocational Institute , Thunder Bay, Ontario , Canada
Shang Yilun
Electronic publication date: 2022 Oct 7
Publication date: 2022
Volume: 8
Electronic Location ID: e1121
Received 2022 May 3; Accepted 2022 Sep 7
Copyright: © 2022 Baxi et al.
Copyright year: 2022
Copyright holder: Baxi et al.
License: This is an open access article distributed under the terms of the Creative Commons Attribution License, which permits unrestricted use, distribution, reproduction and adaptation in any medium and for any purpose provided that it is properly attributed. For attribution, the original author(s), title, publication source (PeerJ Computer Science) and either DOI or URL of the article must be cited.
License URL: https://creativecommons.org/licenses/by/4.0/

Keywords: Social media, Twitter, COVID-19, User engagement, Content analysis, Sentiment strength, Inclusivity and diversity strength, Crisis communication

Funding: Vector Institute, Toronto NSERC Discovery RGPIN-2017-05377 Manmeet Kaur Baxi is funded by an AI Scholarship from Vector Institute, Toronto; an NSERC Discovery Grant (RGPIN-2017-05377) is held by Vijay Mago. The funders had no role in study design, data collection and analysis, decision to publish, or preparation of the manuscript.

==============================
This study assesses the online societal association of leaders and healthcare organizations from the top-10 COVID-19 resilient nations through public engagement, sentiment strength, and inclusivity and diversity strength. After analyzing 173,071 Tweets authored by the leaders and health organizations, our findings indicate that United Arab Emirate’s Prime Minister had the highest online societal association (normalized online societal association: 1.000) followed by the leaders of Canada and Turkey (normalized online societal association: 0.068 and 0.033, respectively); and among the healthcare organizations, the Public Health Agency of Canada was the most impactful (normalized online societal association: 1.000) followed by the healthcare agencies of Turkey and Spain (normalized online societal association: 0.632 and 0.094 respectively). In comparison to healthcare organizations, the leaders displayed a strong awareness of individual factors and generalized their Tweets to a broader audience. The findings also suggest that users prefer accessing social media platforms for information during health emergencies and that leaders and healthcare institutions should realize the potential to use them effectively.

Introduction

The exponential spread of the 2019 severe acute respiratory syndrome coronavirus 2 (SARS-CoV-2) has flooded various social media platforms (SMPs) with a plethora of information about the disease, pandemic trajectory, influence on human fatalities, and global and regional consequences for the governments and health organizations (Gates, 2020). SMPs such as Twitter, Facebook and Instagram have become the norm for broadcasting and acquiring pandemic-related information by leaders and healthcare organizations. Researchers have demonstrated an interest in examining and interpreting the social media data of leaders and healthcare organizations across various SMPs by evaluating their Twitter usage through content analysis and the change in their number of followers (Haman, 2020; Rufai & Bunce, 2020).

Political leaders are followed on Twitter for a number of reasons, including convenience, expressiveness, knowledge and sociability (Parmelee & Bichard, 2011). According to research, 70% of healthcare institutions in the United States also utilize SMPs, and their social media presence influences 57% of clients’ decisions about where to seek medical care (Peck, 2014). Thus, it is necessary to investigate the online societal association of leaders and healthcare organizations on the citizens during a crisis situation. The importance of studying the qualitative factors to determine the online societal association of social media on Government to Citizen interactions has been highlighted previously (Bonsón et al., 2012; Norris & Reddick, 2013). Therefore, the objective of this study is to quantify the online societal association of leaders and healthcare organizations from the top-10 COVID-19 resilient nations by analyzing the following factors–user engagement, sentiment strength, and the inclusivity and diversity of various communities in the Tweets authored by them. Figure 1 depicts the overall research framework.

Figure 1 Overall research framework.

Understanding which information (content, type) appeals to the audience the most might be an effective way of amplifying the involvement (Bonsón, Royo & Ratkai, 2015; Bonsón & Bednárová, 2018), and hence can help in perpetuating a regular conversation with the public, addressing their concerns, recommendations, and desires, and thus assist in pacifying them, building trust, and fighting through crisis situations together. On this rationale, this article calculates the influence of leaders and healthcare organizations, and the engagement they receive during COVID-19. Furthermore, people, especially political leaders and healthcare organizations, communicate their opinions and attitudes–which are generally termed as ‘sentiment’ through several SMPs (Dang-Xuan et al., 2013). However, it is ambiguous how sentiment and the references to different communities might influence information dynamics in a social-media setting. Therefore, it is critical to evaluate the collective influence of sentiment, inclusion and diversity along with the public engagement on the leaders’ and healthcare organizations’ online societal association.

This study offers a thorough assessment of the leaders and healthcare organizations, including both qualitative and quantitative insights regarding their online behaviour. This type of hybrid study has yielded promising findings in the past as well (Sampieri, 2018). Our research examines the various metrics of their Tweets (likes, replies, retweets, and quotes) as well as utilizes statistical methodologies to compute sentiment strength, inclusiveness, and diversity strength. It is qualitative in nature as we analyze the Tweet content to see whether there are any parallels to real-life events and we try to quantify the levels of engagement and online societal association. As a result, the purpose of this study is to explore how audience engagement, sentiment, inclusion and diversity strength may assist leaders and healthcare organizations develop a trustworthy relationship with the public, gathering support for policies that limit the spread of COVID-19, and overcoming the crisis situations. The key findings of the research are: Amongst politicians, United Arab Emirates’ Prime Minister had the highest online societal association,

The Canadian public health agency demonstrated a prominent level of online societal association amid the healthcare organizations, and,

Individual aspects of the online societal association were better understood by leaders than by healthcare organizations, because the latter had lower levels of audience engagement, targeted limited groups, and had relatively low sentiment strength, as seen by the results.

The remainder of the article is structured as follows: Related work section discusses the summary of the past research relevant to the usage of SMPs by leaders and healthcare organizations. Dataset and the statistical methods used to compute the online societal association are discussed in the Methodology section. The results and implications of this study are discussed in Results, followed by the Discussion & Conclusion section, where the key inferences and further research directions are discussed.

Related work

Leaders and healthcare organizations utilize social media platforms (SMPs) including Facebook, Twitter, Instagram, and Reddit for election campaigns, broadcasting public health information, announcing significant events, and improving public relations (Bhattacharya, Srinivasan & Polgreen, 2017; Bertot et al., 2010; Bonsón et al., 2012; Chun et al., 2010).

In lieu of the use of SMPs for the candidate and audience engagement on Twitter, and election campaigns for the 2013 and 2016, Australian federal elections was compared (Bruns & Moon, 2018), while the use of Facebook for strategic campaigning in the 2008 and 2012 US Presidential elections was analyzed (Borah, 2016). Additionally, the use of Instagram as an advertising tool for Swedish elections and YouTube for the electoral campaign of European Parliamentary elections was examined (Russmann & Svensson, 2017; Vesnic-Alujevic & Van Bauwel, 2014). SMPs, according to earlier studies (Bertot et al., 2010; Bonsón et al., 2012; Chun et al., 2010) may assist in improving governance transparency, engagement, and accountability. Furthermore, using a variety of examples, researchers have highlighted the impact on several aspects of public interaction through SMPs (Gruzd & Roy, 2016; Hollebeek, Glynn & Brodie, 2014; Ríos, Benito & Bastida, 2017). These platforms have been cited by several academicians (Bonsón, Perea & Bednárová, 2019; Bonsón & Ratkai, 2013; Bonsón, Royo & Ratkai, 2017; Gruzd, Lannigan & Quigley, 2018; Sahly, Shao & Kwon, 2019; Siebers, Gradus & Grotens, 2019) as an important tool for expanding the social reach and better understanding the audience. However, previous research has also demonstrated that the sentiment, emotion, and diversity of Tweets, can mitigate public engagement and persuade the audience (Bhat et al., 2021; Sandoval-Almazan & Valle-Cruz, 2018; Jünger & Fähnrich, 2020; Qudar, Bhatia & Mago, 2021; Singhal, Baxi & Mago, 2022). Therefore, it is clear that in past few years, political leaders from a variety of nations have actively employed SMPs for both their election campaigns and to comprehend their audience.

As seen recently, SMPs have also been used by healthcare professionals and organizations as a communication tool to promote healthy habits, share announcements, disseminate awareness, motivate the patients on the way to their recovery, support emergency response, and eventually boost readiness during exceptional circumstances (Benetoli, Chen & Aslani, 2018; Househ, 2013; Li & Sakamoto, 2014; Ventola, 2014; Merchant & Lurie, 2020; Shah et al., 2020; Patel et al., 2019, 2020). The idea of SMPs having a positive influence on public awareness and behavioural changes by disseminating succinct information to specified audiences was explored (Al-Dmour et al., 2020). Further, to evaluate public reactions during the epidemic, topic identification and sentiment analysis was utilized to examine the change of public attitude over time in relation to the published news, and reddit posts (Garcia & Berton, 2021; Melton et al., 2021). Also, a previous study has identified factors associated with the levels and duration of engagement, for the Facebook accounts of U.S. Federal health agencies (Bhattacharya, Srinivasan & Polgreen, 2017). Furthermore, researchers have investigated the influence of world leaders during the COVID-19 pandemic and how Twitter was used to swiftly transmit information to the public (Haman, 2020; Rufai & Bunce, 2020). However, there has not been an extensive analysis to understand and examine the online societal association of leaders and healthcare organizations during COVID-19, considering different factors like user engagement, sentiment strength, and, inclusivity and diversity strength, during a health emergency, despite the physical distancing and lockdown measures, which is hence the focus of this work.

Methodology

Dataset

Twitter Academic Research API v2 (https://developer.twitter.com/en/products/twitter-api/academic-research) was utilized to retrieve the information of the political leaders’ and health organizations’ Tweets. A total of 173,071 Tweets were collected and analyzed from December 1, 2019, to December 31, 2021. The dataset was curated based on the Bloomberg COVID-19 Resilience Ranking (https://www.bloomberg.com/graphics/covid-resilience-ranking/), as of January 8, 2022, at 5 p.m. EST, selecting the health organizations and leaders of the top-10 COVID-19 resilient countries. The COVID-19 Resilience Ranking is a monthly impression of the countries handling the virus most effectively, with the least social and economic disruption. The ranking is calculated based on the factors of virus containment, quality of healthcare, vaccination coverage, overall mortality and progress towards restarting travel. The timeline was chosen to include the outbreak COVID-19 to the vaccination period of the pandemic. Official health organizations of the respective countries and personal accounts of the political leaders were analyzed in this specific study. This provides an opportunity to get a sense of the contrasting dynamics between the accounts; to truly encapsulate the online societal association on the particular country.

The collected Tweets spanned across 19 different languages and were translated to English using the Neural Machine Translation (NMT) models from the Tatoeba Translation Challenge, which consists of NMT models trained on a compressed dataset of over 500 GB, encompassing 2,961 language pairings, and 555 languages (Tiedemann, 2020). For this study, each Twitter account is referred to as a user and the type of account (i.e., leaders, health organizations) is referred to as a user group. The details of the Tweets authored by each of the selected users in the order of their COVID-19 Resilience ranking (i.e., from the best country to live in during COVID-19, like Chile, to the good ones, like United Kingdom) can be found in Table 1.

Table 1 Distribution of Tweets for the selected user accounts.

Account type	Name (Twitter handle)	Country	Number of Tweets	
Leader (President or Prime Minister)	Sebastián Piera (@sebastianpinera) (President)	Chile	622	
	Micheál Martin (@PresidentIRL, @MichealMartinTD)	Ireland	3,641	
	Mohammed bin Rashid Al Maktoum (@HHShkMohd)	U.A.E	839	
	Sanna Marin (@MarinSanna)	Finland	2,007	
	Justin Trudeau (@JustinTrudeau, @CanadianPM)	Canada	13,778	
	Iván Duque (@IvanDuque) (President)	Colombia	7,059	
	Recep Tayyip Erdoğan (@RTErdogan) (President)	Turkey	1,943	
	Pedro Sánchez (@sanchezcastejon)	Spain	4,290	
	Magdalena Andersson (@SwedishPM)	Sweden	282	
	Boris Johnson (@BorisJohnson)	United Kingdom	2,335	
	Total	36,796	
Health Organization/Health Minister	Ministerio de Salud (@ministeriosalud)	Chile	39,401	
	HSELive (@HSELive), Department of Health (@roinnslainte)	Ireland	18,332	
	Ministry of Health and Prevention, U.A.E. (@mohapuae)	U.A.E	8,424	
	Ministry of Social Affairs and Health (@MSAH_News)	Finland	1,009	
	Health Canada and PHAC (@GovCanHealth)	Canada	38,715	
	Ministry of Health and Social Protection of Colombia (@MinSaludCol)	Colombia	11,346	
	Ministry of Health of the Republic of Turkey (@saglikbakanligi)	Turkey	4,119	
	Ministry of Health (@sanidadgob)	Spain	8,595	
	The Public Health Agency of Sweden (@Folkhalsomynd)	Sweden	701	
	UK Health Security Agency (@UKHSA)	United Kingdom	5,633	
	Total	136,725	

Online societal association

The online societal association (denoted by, onlineSocietalAssociation) is defined as the product of engagement per day with user impact (dailyAvgEnguser), sentiment strength (sentiStrength), and inclusivity and diversity strength (iDStrength) in the user’s Tweets as in Eq. (1). Each criterion is given the same weight since they have all been scaled to account for bias, and all of these parameters come together to generate a Tweet which is addressed to the general audience amongst the network of healthcare organizations and leaders of the top-10 COVID-19 resilient countries in the Twitter ecosystem. Further details of the variables can be found in the following subsections.

(1) onlineSocietalAssociation=dailyAvgEnguser∗sentiStrength∗iDStrength

Engagement with impact

The engagement per day is the measure of the social interaction of the post, including the likes, replies, retweets and quotes. The engagement per day represents the relationship between the followers and the user, and the resonation of their Tweets. Twitter defines engagement rate, as the ratio of engagements to impressions: EngagementImpressions×100. The engagements are defined as an aggregate of interactions of a Tweet–retweets, replies, follows, likes, links, cards, hashtags, embedded media, profile photo, username or Tweet expansion. The impressions account for times a user has observed a particular Tweet in their search results or timeline (Twitter Account Activity Analytics–Engagement, Impressions) (https://help.twitter.com/en/managing-your-account/using-the-Tweet-activity-dashboard). This study analyzes only public metrics such as the count of likes, replies, retweets and quotes-as a result of the limitations of Twitter API.

To evaluate a user’s engagement (dailyAvgEnguser, Eq. (2)); firstly, their Tweet-wise engagement (dailyAvgEng(Tweet,user), Eq. (3)) is calculated by multiplying the user impact (impactuser) and average engagement per day for a Tweet, (dailyAvgEngTweet), followed by taking an average of the Tweet-wise engagement (dailyAvgEng(Tweet,user)) for the user.

(2) dailyAvgEnguser=∑dailyAvgEng(tweet,user)totalTweetsuser

(3) dailyAvgEng(tweet,user)=dailyAvgEngtweet×impactuser

The impact of a user (impactuser, Eq. (4)) is quantified based upon the hyperbolic tangent function (tanh) of followers, the total number of Tweets, following, public lists and profile age. The listedCount is the total amount of public lists of a user. Lists indicate popularity–generally revolving around the concept that other users are engaged with one’s content. Furthermore, log10(followersfollowing) is the followers to following ratio, indicating the general nature of the account. The ratio is within a base-10 log to elude outlier values. The totalTweetCount is the number of Tweets from the account during our data collection timeframe. The profileAge represents the number of days between the profile creation date to December 31, 2021; the last analysis day. Because a freshly joined user with more followers would be more influential than a previously joined user with fewer followers, the square of a user’s profile age has been deemed inversely proportional to the user’s impact.

(4) impactuser=tanh⁡(log10⁡(followersfollowing)×listedCount×tweetCount)(profileAge)2

To quantify the average engagement per day (dailyAvgEngTweet, Eq. (5)), the collated number of likes, replies, retweets, quotes and Tweets per day, from December 1, 2019, to December 31, 2021, is used. Furthermore, the dailyTweetCount are multiplied by 4 (equal to the number of variables in the numerator).

(5) dailyAvgEngtweet=likes+replies+retweets+quotes4∗dailyTweetCount

To standardize the shifting values of average engagement, we calculate the Exponential Moving Average with a 151-day window span1 , eliminate outliers using z − score and smoothen the average engagement per day to the 8th degree using the Savitzky Golay filter2 .

Sentiment strength

To quantify the strength of sentiment for every user, we first calculate the sentiments of all the Tweets collected for our analysis using CardiffNLP’s ‘twitter-roberta-base-sentiment’ model, which is trained on a 60 million Twitter corpus, and then calculate the sentiment strength for every user as mentioned in Eq. (6), i.e., based on the sentiment category with the maximum number of Tweets for that day, followed by assigning the sentiment score based on the sentiment: 10−6 for neutral, the ratio of the count of positive Tweets to total Tweets for positive, and the negation of the ratio of the count of negative Tweets to the total Tweets for negative sentiment.

(6) sentiStrength={10−6;maxSentimentScore(tweets)=neutralcount(positiveTweets)totalTweets;maxSentimentScore(tweets)=positive−count(negativeTweets)totalTweets;maxSentimentScore(tweets)=negative

Inclusivity and diversity strength

We assessed the inclusivity and diversity in the Tweets of the users (denoted by, iDStrength) by computing the usage of different keywords pertaining to various communities from the countries selected for our analysis. The keywords were selected based on gender, age, cultural inferences, ethnicity, and employment sectors of each of these countries. The detailed list of keywords can be found in the GitHub repository (https://github.com/manmeetkaurbaxi/Societal-Impact-on-Twitter). The usage frequency for each of these keywords is calculated for all users with respect to the total number of Tweets from that user, as given in Eq. (7). If there exists a user who has not referred to any community in their Tweets, a default value of 10−6 is assigned.

(7) iDStrength={      10−6    ;count(communityMentionTweets)=0count(communityMentionTweets)totalTweets;       otherwise

Content analysis

The Tweets of all the users were analyzed for the most-frequent topics and the most-referred users by assessing the usage of hashtags and mentions. The Tweets were examined by extracting the top-10 hashtags and mentions using the ‘advertools 0.13.0’ module (https://pypi.org/project/advertools/). We compare the similarities and differences in the tweeting habits of health organizations and leaders of the top-10 COVID-19 resilient countries.

Computational resources and GitHub

The analysis was done using the Digital Research Alliance of Canada’s service (https://www.computecanada.ca/research-portal/accessing-resources/available-resources/). The computational resources provided by the ‘graham’ cluster of the Digital Research Alliance of Canada were as listed below: CPU: 2× Intel E5-2683 v4 Broadwell @ 2.1 GHz

Memory (RAM): 30 GB

The supplementary material for this study–data, code, and results are available on the GitHub repository (https://github.com/manmeetkaurbaxi/Societal-Impact-on-Twitter).

Results

Online societal association

Prime Minister of the U.A.E, Mohammed bin Rashid Al Maktoum (online societal association: 1.000), had the highest online societal association overall, followed by Canadian Prime Minister Justin Trudeau (online societal association: 0.068) and Turkish President Recep Tayyip Erdoğan (online societal association: 0.033), among the leaders (Fig. 2A). Out of the health organizations (refer Fig. 2B), the Health Canada and Public Health Agency of Canada (PHAC, online societal association: 1.000) had the highest online societal association, followed by the Ministry of Health of the Republic of Turkey (online societal association: 0.632) and the Ministry of Health of Spain (online societal association: 0.094). The results for each of the factors affecting the online societal association, are individually explained in the following subsections.

Figure 2 Online societal association (scaled) of the top-5 most impactful users by their user group, i.e., (A) health organizations, and (B) leaders of the top-10 COVID-19 resilient countries.

Engagement with impact

The user impact was scaled between the range 0 and 1 (1 denotes high user impact, and 0 denotes low user impact). The results indicate that the Turkish President (Recep Tayyip Erdoğan) had the greatest user impact (1.000), followed by U.K. Prime Minister (Boris Johnson, user impact: 0.978), and the Prime Minister of U.A.E (Mohammed bin Rashid Al Maktoum, user impact: 0.663) among the leaders. Among the health organizations, the Ministry of Health of the Republic of Turkey had the highest user impact (1.000), followed by the Ministry of Health and Social Protection of Colombia (user impact: 0.887) and the UK Health Security Agency (user impact: 0.778).

Among the health organizations, The Ministry of Health of the Republic of Turkey’s user engagement is considerably higher than the other organizations (Fig. 3A). The highest engagement was observed during April, 2020. This can be attributed to the impacts of COVID-19, specifically, the curfew mandate imposed by the Turkish government during this time. The user engagement gradually decreased, as the COVID-19 measures lifted, and the normalization process continued. Similar to the health organizations, Turkish President Recep Tayyip Erdoğan’s user engagement (as shown in Fig. 3B) is considerably higher than the other leaders, with the highest engagement recorded during August–October, 2020. The initial rise in engagement came in response to the finding of 320 billion cubic metres of natural gas in the Black Sea, which was made possible by drilling in the Danube-1 well, which began on July 20, 2020, as part of their goal of being a massive energy exporter (Cohen, 2020). The subsequent spike in engagement occurred in the aftermath of the 6.6 magnitude earthquake that struck Izmir, Turkey on October 30, 2020, with the government agencies rallying to save people who were trapped (OCHA, 2020).

Figure 3 Average engagement per day with user impact for (A) health organizations, and (B) leaders ofthe top-10 COVID-19 resilient countries.

Sentiment strength

After computing the Sentiment Strength for all the users, it was found that most of the users had a neutral outlook on Twitter, except for the UK Health Security Agency (UKHSA), who had a highly negative opinion (sentiment strength: −0.999). Only six user accounts out of 20 reflected positive sentiment through their Tweets; five of these were the leaders of Chile, U.A.E., Canada, Colombia, Sweden (as shown in Fig. 4) and the official account of the Public Health Agency of Canada (PHAC) (sentiment strength: 0.411). Among the leaders, U.A.E.’s Prime Minister Mohammed bin Rashid Al Maktoum had the highest positive sentiment strength (i.e., 0.746), followed by the Swedish Prime Minister, Magdalena Andersson (sentiment strength: 0.706) and Canadian Prime Minister, Justin Trudeau (sentiment strength: 0.512). Figure 4 depicts the sentiment strength of the top-5 leaders.

Figure 4 Sentiment strength of top-5 leaders.

Inclusivity and diversity strength

Leaders of the top-10 COVID-19 resilient countries were more inclusive and diverse while tweeting compared to the health organizations of these countries. Among the leaders, U.A.E’s Prime Minister had the highest inclusivity and diversity strength (i.e., 0.644), followed by the Colombian President, Iván Duque (inclusivity and diversity strength: 0.63), and the Chilean President, Sebastián Piñera (inclusivity and diversity strength: 0.624). For the health organizations, the results were slightly different, with Finland’s Ministry of Social Affairs and Health having the highest inclusivity and diversity strength (i.e., 0.534), followed by the Colombian Ministry of Health and Social Protection (inclusivity and diversity strength: 0.407) and U.A.E.’s health organization, MOHAP (inclusivity and diversity strength: 0.303). Figures 5A and 5B illustrates the Inclusivity and Diversity Strength of the top-5 health organizations and leaders respectively.

Figure 5 Inclusivity and diversity strength for top-5 (A) health organizations, and (B) leaders.

Content analysis

The analysis of the hashtags served as a representation of the content, for each of the users. ‘COVID-19’ was the most discussed topic among the accounts of the health organizations-as shown in Fig. 6A which displays the high frequency of ‘#covid19’, ‘#covid_19’, ‘#yomevacuno’ (referring to the vaccination plans and status of Chile) (https://www.gob.cl/yomevacuno/), and ‘#coronavirus’ hashtags in user’s Tweets. The data indicates that the health organizations communicated about the COVID-19 pandemic with the same, or similar words. However, the results of the political leaders indicated contrasting content discussion. This is due to the fact that the political leaders would discuss relevant political issues, respective to their country-attributing to the diversity of hashtags. This concept is supported by Fig. 6B which shows the variety of hashtags related to different topics; ‘#covid19’, ‘#cop25’ (referencing the 25th United Nations Climate Change Conference, held from December 2 to 13, 2019 (https://unfccc.int/cop25)), ‘#tokyo2020’, ‘#budget2021’, ‘#euco’ (regarding the European Council (https://www.consilium.europa.eu/en/european-council/)), ‘#Bogotá’ (referring to the capital of Columbia), ‘#Fuerzaública’ (referencing the public force of Columbia (https://www.constitucioncolombia.com/titulo-7/capitulo-7)), and ‘#machnamh100’ (regarding an initiative of Ireland’s President, Michael D. Higgins, which explores influential events during Ireland’s Decade of Commemorations (https://president.ie/en/news/article/machnamh-100-president-of-irelands-centenary-reflections)). The frequency of mentions were measured to understand the interactions of each account. Figure 7A indicates the frequency of mentions among the health organizations. The mentioned accounts are current or former health ministers, or politicians, respective to each of the top-10 COVID-19 resilient countries selected for our analysis. Figure 7B represents the most recurrent mentions for the political leaders. The mentioned accounts were relevant political figures or organizations to the country.

Figure 6 Top #tags for (A) health organizations, and (B) leaders of the top-10 COVID-19 resilient countries.

Figure 7 Top mentions for (A) health organizations, and (B) leaders of the top-10 COVID-19 resilient countries.

Discussion and Conclusion

Principal findings

Through this research, we proposed a framework for extensive analysis of social media content and the online societal associations of healthcare organizations and leaders of the top-10 COVID-19 resilient nations using NLP-based text-mining and statistical methodologies. We evaluated reasonably significant amounts of textual data for assessing the online societal association by analyzing impact and engagement, sentiment strength, and inclusion and diversity strength. The significant conclusions from our research are as follows: Being the most active user on social media does not necessarily imply a higher level of online societal association. The Prime Ministers of the United Arab Emirates and Canada had significantly more online societal association than the leaders of Colombia and Spain, despite the latter’s having a higher number of Tweets. A similar observation is made for the health organizations, where the Canadian and Turkish health agencies created a substantially more significant online societal association than Colombia and Ireland. According to our findings, people are also more inclined to engage with neutral Tweets, which normally contain some sort of public notification, rather than entirely positive or negative Tweets. This might imply that healthcare organizations and leaders can use this information to their advantage when developing content for social media postings to maximize their online societal associations.

Using specific hashtags undoubtedly aids in driving engagement, as we have seen that most public engagement is highly slanted towards Tweets containing hashtags related to ‘COVID-19’. Furthermore, we note that user engagement for both the user groups, i.e., health organizations and leaders, follows a predictable pattern, with peaks emerging around events of emergency or public welfare announcements (For instance, there was more public interaction when the Turkish President announced the discovery of 320 billion cubic metres of natural gas in the Black Sea during August–October 2020 and when the Ministry of Health of the Republic of Turkey tweeted about the curfew mandate in April 2020).

Leaders of the top-10 COVID-19 resilient nations targeted wider audiences than their health organizations when it came to inclusion and diversity. Additionally, they portrayed a comparably higher sentiment strength from the health organizations.

Limitations and future work

The results of this study are confined to the COVID-19 timeline selected, i.e., between December 1, 2019, to December 31, 2021. To further comprehend the online societal association of leaders and health organizations in different timeframes, the researchers might use alternative approaches to organize their data. Moreover, our research focuses on leaders’ and healthcare organizations’ Twitter data, which is often clean and requires little pre-processing. Because our research was confined to textual data, we could not account for the influence of image characteristics or knowledge graphs related to individual Tweets. However, it would be intriguing to see how this methodology behaves on the Tweets of other cabinet members and decision-makers of these countries, as well as investigate the organic and paid audiences, if any exist. Another area that future research might look at is the demographics of the individuals who are interacting with these contents and the penetration of Twitter among different countries.

Conclusion

This study investigated the online activity of healthcare organizations and leaders of the top 10 COVID-19 resilient nations on Twitter. The NLP-based statistical methods analysis of the social media activity presented here can be utilized to gauge the online societal association on the previously published Tweets and to generate Tweets that create an impact on people accessing healthcare information via SMPs. Each individual characteristic, i.e., public impact and involvement, sentiment strength, inclusivity and diversity strength, played an equally important role in determining a user’s online societal association. Thus, we believe that quantifying the online societal association and analyzing the Tweet content provides a better understanding of how posting the appropriate Tweet at the right time may make all the difference in society.

The authors would like to express their gratitude to Digital Research Alliance of Canada and CASES building, Lakehead University for providing the computational resources needed to complete this research, and DaTALab member Aditya Singhal for proof-reading the manuscript.

Additional Information and Declarations

Competing Interests

Author Contributions

Data Availability

1 A grid-search analysis was performed to find the best value.

2 A grid-search analysis was performed to find the best value.

Vijay Mago is an Associate Editor for IEEE Access and BMC Medical Informatics and Decision Making, and an Academic Editor for PeerJ.

Manmeet Kaur Baxi conceived and designed the experiments, performed the experiments, analyzed the data, performed the computation work, prepared figures and/or tables, authored or reviewed drafts of the article, and approved the final draft.

Joshua Philip conceived and designed the experiments, authored or reviewed drafts of the article, and approved the final draft.

Vijay Mago conceived and designed the experiments, authored or reviewed drafts of the article, and approved the final draft.

The following information was supplied regarding data availability:

The code and data are available at GitHub: https://github.com/manmeetkaurbaxi/Societal-Impact-on-Twitter.

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
