# Peer review of "Resilience of political leaders and healthcare organizations during COVID-19"

_PeerJ Computer Science, doi:10.7717/peerj-cs.1121_

## Round 0.1 · original submission · Major Revisions

We have received three detailed reports for the paper. They have pointed some concerns in design and results. Please revise the paper and provide a detailed response letter. Thanks.

·

Basic reporting

In overall, I find the manuscript very clearly written in English. Through public involvement, sentiment strength, and inclusivity and diversity strength, the author highlighted the societal impact of leaders and healthcare organizations from the top-10 COVID-19 resilient nations. This research provides a comprehensive examination of healthcare leaders and organizations, including 58 qualitative and quantitative insights into their online behavior. For election campaigns, disseminating public health information, announcing 80 key events, and strengthening public relations, leaders and healthcare organizations use social media platforms (SMPs) such as Facebook, Twitter, 79 Instagram, and Reddit. The author used the method twitter academic research API v21 and Sentiment Strength to retrieve information from 117 health groups and political leaders. Discussion section is also good which is described by the author.

Experimental design

The paper describes an experimental advance in the field of epidemic Field and social media responsibility. This manuscript is also described in excellent way about Societal Impact, Engagement with impact, Sentiment Strength and Inclusivity and Diversity Strength. Among the leaders, Prime Minister of the United Arab Emirates (societal impact: 1.000), had the highest 195 societal impact, followed by Canadian Prime Minister (societal impact: 0.068) and Turkish President (societal impact: 0.033). Health Canada and the Public Health Agency of Canada (PHAC, societal impact: 1.000) 198 had the highest societal impact of the 197 health organizations, followed by the Ministry of Health of the Republic of Turkey (societal impact: 0.632) and the Ministry of Health of Spain (societal impact: 0.632). (Societal impact: 0.094). The authors of this study use NLP-based text mining algorithms to evaluate the societal impact of leaders and health organizations from the top-10 COVID-19 resilient countries. We evaluated large amounts of textual data in order to estimate the societal impact by gauging public involvement, sentiment strength, and inclusivity and diversity strength. The data also show that being the most active user on social media does not always suggest a better level of societal effect, according to the authors.

Validity of the findings

The findings and recommendations by authors are not new, and there have been prior incremental moves to expand what constitutes, influences and governs health and healthcare. COVID-19 demands dynamic systemic transformation. The pandemic has fundamentally challenged health systems and the communities they serve globally. The effect of a major shock represented by the pandemic is to manifest the points where the system is weakest, and to demonstrate the interdependencies of a range of health, social and economic structures. While the evidence of system failures has come at a huge cost in human and monetary terms, it has also pointed to what needs to change. Thus, we have found out that the manuscript is valid and relent for journal publications. Dataset is also valid and relevant for such type of research.

Additional comments

1. It would be better to rewrite the whole manuscript and highlight the main aim with contributions in it. Also, quantitative analysis is missing.
2. The Introduction section is very short and trivial. I suggest to increase its length and also at the end of this section write down the contributions pointwise. The novelty and objective of the investigation compared to published articles should be mentioned.
3. The related work or Literature Review section must should have in this paper which will highlight the main issues of the recent work related to this paper.
4. Illustration presented in different figures are not well formatted. Most of them are blurred image, hard to recognize its contents.
5. Experimental Results (if any) should incorporated after implementation. Need discussion with comparative analysis with the existing solutions for improving the manuscript.
6. The proposed architecture shall be explained in a better manner. It must be crisp and clear for the readers. The author may use flowchart for the same, additionally.
7. Conclusion is important part of a research paper. The explanation made in conclusion is not doing justice with the work done. This section should be revised.

·

Basic reporting

1) There are several language and clarity issues throughout, such as incorrect use of articles and tenses. This phrase, for example, is very unclear: "After computing the Sentiment Strength for all the users, it was found that most of the users had a neutral outlook on Twitter, except for the UK Health Security Agency (UKHSA), who had a highly negative opinion (sentiment strength: −0.999)." I’d recommend getting professional revisions from a colleague.

2) The relevance of literature references are insufficiently explained (for example, the majority of references do not explain the relevance of a given finding).

Experimental design

1) I'm very unconvinced that the measure of societal impact is justified. It needs to be explained far more clearly why (daily engagement * sentiment strength * inclusivity and diversity strength) is a good measure for societal impact. e.g. why they should be weighted equally. This needs to be justified in a lot more detail, or the data should be reanalysed, if, as I suspect, it may not be justifiable.

2) The authors state that "the purpose of this study is to explore how audience engagement, sentiment, inclusion and diversity strength may assist leaders and healthcare organizations develop a trustworthy relationship with the public, gathering support for policies that limit the spread of COVID-19, and overcoming the crisis situations together". These questions are not responded to in the conclusion, and I don’t believe they are sufficiently addressed in the study.

Validity of the findings

1) Conclusions are generally overstated. One key finding is: "Individual aspects of the societal impact were better understood by leaders than by healthcare organisations" does not appear to be justified, as there is no measurement of 'understanding aspects of societal impact' in the study design.

2) There has been no attempt to control for national variables (twitter use in different countries may vary greatly etc.)

3) Some of the conclusions are effectively restating the premises: "As a result, each of the individual characteristics, i.e., public involvement, sentiment strength, inclusivity and diversity strength, played an equally important role in determining a user’s societal influence."

Reviewer 3 ·

Basic reporting

The language used is clear and professional, which makes the paper easy to read and understand. The structure of the paper is adequate but there is one part that in my opinion should appear elsewhere. In the introduction, from line 66 to 77, the main conclusions are discussed and this should be in the summary (which is already established) and in the conclusions (where it also appears) but not in the introduction. What is fine is what follows after this paragraph, which is the structure of the research paper once we have set the main objective.
The background is well justified, although there is little literature on the subject analysed.
It presents quality tables and graphs that illustrate the text and give complementary information and is well labelled.
I would recommend checking that the citations in the articles are all in line with the journal's guidelines.
The references are current and sufficient to support this work.

Experimental design

No comment

Validity of the findings

This paper highlights the importance of understanding how social media platforms work. Knowing that the effectiveness of our communication campaigns does not depend on the degree of activity we have but is more linked to the content of what we publish is a first step in this area of new technologies.
In addition, this work provides us with information on what the content should be like if we want it to have the desired effectiveness, neutral messages, neither positive nor negative, and which are aimed at a broad target audience.
This can also be applied to many other areas, not only health, but also political, social or business.

Additional comments

This is a work that covers a very little explored topic and which the authors carry out in an appropriate manner, following a typical structure for this type of work. To do so, it uses up-to-date bibliography related to the subject and also obtains information that is analysed in order to reach conclusions that add value to existing knowledge. This work highlights the importance of knowing how social media platforms work. Knowing that the effectiveness of our communication campaigns does not depend on the degree of activity we have, but is more linked to the content of what we publish, is a first step in this area of new technologies.
These results can also be contrasted in many other areas, not only in healthcare, but also in political, social and business areas.

---

## Round 0.2 · accepted · Accept

The authors have addressed the reviewers' comments and it is now in a good shape. I recommend it for publication.

·

Basic reporting

In overall, I find the manuscript very clearly written in English. Through public involvement, sentiment strength, and inclusivity and diversity strength, the author highlighted the societal impact of leaders and healthcare organizations from the top-10 COVID-19 resilient nations. Experimental and Discussion section is also good which is described by the author.

Experimental design

The paper describes an experimental advance in the field of epidemic Field and social media responsibility. This manuscript is also described in excellent way about Societal Impact, Engagement with impact, Sentiment Strength and Inclusivity and Diversity Strength.

Validity of the findings

The findings and recommendations by authors are not new, and there have been prior incremental moves to expand what constitutes, influences and governs health and healthcare. COVID-19 demands dynamic systemic transformation. The pandemic has fundamentally challenged health systems and the communities they serve globally. The effect of a major shock represented by the pandemic is to manifest the points where the system is weakest, and to demonstrate the interdependencies of a range of health, social and economic structures. While the evidence of system failures has come at a huge cost in human and monetary terms, it has also pointed to what needs to change. Thus, we have found out that the manuscript is valid and relent for journal publications. Dataset is also valid and relevant for such type of research.

Additional comments

No additional Comments